# Predictive Value of [^99m^Tc]-MAA-Based Dosimetry in Hepatocellular Carcinoma Patients Treated with [^90^Y]-TARE: A Single-Center Experience

**DOI:** 10.3390/diagnostics13142432

**Published:** 2023-07-20

**Authors:** Michael Doppler, Marlene Reincke, Dominik Bettinger, Katharina Vogt, Jakob Weiss, Michael Schultheiss, Wibke Uller, Niklas Verloh, Christian Goetz

**Affiliations:** 1Department of Diagnostic and Interventional Radiology, Medical Center University of Freiburg, Faculty of Medicine, University of Freiburg, 79085 Freiburg, Germany; 2Department of Medicine II, Medical Center University of Freiburg, Faculty of Medicine, University of Freiburg, 79085 Freiburg, Germany; 3Berta-Ottenstein-Programme, Faculty of Medicine, University of Freiburg, 79085 Freiburg, Germany; 4Department of Nuclear Medicine, Medical Center University of Freiburg, Faculty of Medicine, University of Freiburg, 79085 Freiburg, Germany

**Keywords:** TARE, radioembolization, ^90^Y-dosimetry, hepatocellular carcinoma, retrospective analysis

## Abstract

Transarterial radioembolization is a well-established method for the treatment of hepatocellular carcinoma. The tolerability and incidence of hepatic decompensation are related to the doses delivered to the tumor and healthy liver. This retrospective study was performed at our center to evaluate whether tumor- and healthy-liver-absorbed dose levels in TARE are predictive of tumor response according to the mRECIST 1.1 criteria and overall survival. One hundred and six patients with hepatocellular carcinoma were treated with [^90^Y]-loaded resin microspheres and completed the follow-up. The dose delivered to each compartment was calculated using a compartmental model. The model was based on [^99m^Tc]-labelled albumin aggregate images obtained before the start of therapy. Tumor response was assessed after three months of treatment. Kaplan-Meier analysis was used to assess survival. The mean age of our population was 66 ± 13 years with a majority being BCLC B tumors. Forty-two patients presented with portal vein thrombosis. The response rate was 57% in the overall population and 59% in patients with thrombosis. Target-to-background (TBR) values measured on initial [^99m^Tc]MAA-SPECT-imaging and tumor model dosimetric values were associated with tumor response (*p* < 0.001 and *p* = 0.009, respectively). A dosimetric threshold of 136.5 Gy was predictive of tumor response with a sensitivity of 84.2% and specificity of 89.4%. Overall survival was 24.1 months [IQR 13.1–36.4] for patients who responded to treatment compared to 10.4 months [IQR 6.3–15.9] for the remaining patients (*p* = 0.022). In this cohort, the initial [^99m^Tc]MAA imaging is predictive of response and survival. The dosimetry prior to the application of TARE can be used for treatment planning and our results also suggest that the therapy is well-tolerated. In particular, hepatic decompensation can be predicted even in the presence of PVT.

## 1. Introduction

Transarterial radioembolization (TARE) is an established treatment for patients with malignant liver tumors, particularly in primary liver disease. In hepatocellular carcinoma (HCC), TARE is used as an alternative to transarterial chemoembolization (TACE) or percutaneous radiotherapy in intermediate and advanced stages according to the Barcelona liver cancer group classification (BCLC), as well as according to the German national guidelines [1,2]. TARE can also be used as a bridging therapy to transplantation or resection [2]. The anti-tumor effect of TARE is mainly obtained through the radiation deposited after β-emission from the yttrium-90 nuclides [^90^Y] encapsulated in glass or resin microspheres. These beads are applied directly to the hepatic arterial vasculature, but, because of the small particle size, the macroembolic effect is not as pronounced as with TACE, and TARE has been shown to be safe in patients with portal vein thrombosis (PVT) [1,3]. Tolerance to the therapy is also reasonable compared to alternatives such as percutaneous radiotherapy or chemotherapy, with good quality of life reported during and after treatment [4].

In terms of overall survival (OS), the various randomized controlled trials of TARE in HCC patients have so far failed to demonstrate superiority to sorafenib [5,6] or TACE [7]. The main challenges remain patient selection, the order of application of this therapy in the proposed setting, and, most importantly, dosimetry. The latter has been a particular focus in recent years as the dosimetric approaches used to date have been suspected of being insufficient to achieve a good tumor response while limiting treatment-related toxicity [8]. Increasingly, personalized dosimetric approaches are being proposed with more selective applications and aiming at sparing as much healthy tissue as possible while targeting the tumor volume more efficiently [8,9,10,11,12,13,14,15] in order to improve the efficacy of TARE, which needs to be confirmed by prospective studies that have been initiated [16].

In this single-center retrospective study, we aimed to investigate the outcome of patients with intermediate and advanced HCC treated with TARE, and to confirm the link between tumor dosimetry and response to therapy in this cohort. We also aimed to establish the same relationship between healthy tissue irradiation and the occurrence of liver complications [17]. In our dosimetric approach, we have chosen to use data from the liver perfusion macroaggregated albumin (MAA) scan—the only examination performed prior to TARE administration and, therefore, the only one suitable for therapy planning [11,18,19].

## 2. Materials and Methods

### 2.1. Patient Selection and Follow-Up

One hundred nineteen patients with HCC were consecutively treated with TARE between January 2015 and February 2023 at the University Hospital Freiburg and included in this observational study. The therapy was recused in six further patients during this period due to excessive hepatopulmonary shunting. All treated patients had exclusive or dominant liver disease and HCC was diagnosed based on typical radiological or histological features, according to current EASL and AASLD recommendations [20,21]. The tumor stage was assessed by BCLC [22].

All included patients were scheduled for treatment after discussion in the multidisciplinary tumor board. TARE was decided to be the most suitable first-line therapy for 76 patients. In 41 patients, therapy was recommended because of tumor recurrence; in 23 patients recurrence occurred after initial liver resection, in 15 after TACE, and in 3 of them after percutaneous radiotherapy. Two patients were treated while waiting for a liver transplant (bridging therapy). A minimum delay of three months was observed between applying TARE and any prior therapy (surgery, TACE, or radiotherapy). Portal vein thrombosis was not an exclusion criterion for TARE. The inclusion scheme is summarized in Figure 1 and the detailed characteristics of the included patients are shown in Table 1.

Before application and as part of the initial workup, diagnostic imaging was performed, consisting of a multiphase CT scan with a contrast agent (Ultravist, Bayer Vital GmbH, Leverkusen, Germany). According to the recommendations and manufacturer specifications, all patients were required to have bilirubin ≤2 mg/dL and tumor replacement <50% of total liver volume on diagnostic imaging [19,23].

Patients lost to follow-up at three months post-irradiation were excluded from the final analysis (*n* = 13 patients). The demographic data of the remaining 106 patients were collected at baseline. Clinical, laboratory, and treatment-related data were collected at different time points (at least, baseline, four to six weeks, and three months after the last treatment—as well as during the oncological follow-up, if applicable). The radiological response was assessed three months after TARE (after the second lobar treatment, if applicable).

### 2.2. Pre-Administration Workup Angiography and Simulation

During the workup process, all patients underwent diagnostic abdominal angiography. If necessary, prophylactic embolization of the gastroduodenal artery and the right gastric artery and its pancreaticoduodenal branches was performed and the optimal position of intrahepatic therapy was determined, following practice standards and manufacturer’s instructions [19,23,24,25].

At the end of the angiographic procedure, 135 ± 13 MBq of technetium-99m macroaggregated albumin ([^99m^Tc]MAA) in 5 mL of physiological solution was slowly injected through the arterial microcatheter as a surrogate to simulate the delivery of the therapeutic agent [25,26]. Per session, patients received their infusion from a single location sufficient to cover the tumors. The delivery position was recorded on the angiogram as a reference to allow accurate catheter repositioning for the upcoming treatment session. Subsequent planar and tomographic acquisitions ([^99m^Tc]MAA scintigraphy and SPECT) were performed within one hour and used as previously described to assess the physiological lung or aberrant intra-abdominal shunt [19,24,25,26,27]. Whole-body planar images were acquired in the supine position (15 cm/min scans, low-energy high-resolution collimators, window, 140 +/− 7.5 keV, zoom = 1, 1024 × 256 pixel matrix) on a Siemens Intevo camera (Siemens Symbia Intevo 6, Siemens Healthcare GmbH, Erlangen, Germany). Abdominal SPECT/CT images were acquired with the same camera (128 projections, 20 sec/project) and reconstructed using a 3D-OSEM algorithm (5 iterations, 8 subsets) with a Butterworth filter (cut-off, 0.5 cycles/cm; order 10), CT-based attenuation correction, and diffusion correction on a Syngo workstation (Siemens Healthcare GmbH, Erlangen, Germany). Whole-body planar imaging was carefully reviewed and processed for liver–lung shunt calculation and the SPECT/CT dataset was analyzed to exclude any relevant gastrointestinal shunt. The liver and lung regions of interest and the geometric mean method were applied to whole-body planar images acquired using PMOD (version 3.8; PMOD Technologies Ltd., Zurich, Switzerland), as recommended [19,24,25,26,27,28,29]. Any hepato-pulmonary shunt fraction greater than 20% or significant gastrointestinal arterial reflux without the possibility of additional coiling excluded patients from therapy.

### 2.3. TARE—Application and Post-Therapeutic Imaging

TARE application or, at least, the first application of the microspheres in the case of bilateral therapies was performed within eight days of the workup and the yttrium activity to be injected was calculated conventionally using the manufacturer’s recommended standard body surface area (BSA) method [25,26] until January 2021 and assuming a uniform distribution of yttrium activity in the liver tissue. The activity to be implanted was adjusted according to the tumor size in the treated part of the liver and the size of the patient [25,26,28].

From February 2021 onwards, activities were planned using the partition model for the last 28 patients of our cohort according to the new recommendations in use in our institution and following the international recommendations for personalized TARE [19]. The TARE application used exclusively [^90^Y] resin microspheres (Sirtex Medical Europe GmbH, Bonn, Germany). A sequential approach, as recommended [13,29,30,31], was preferred, avoiding the main hepatic artery whenever possible. When lesions were limited to one lobe, the catheter was selectively inserted into the left or right lobar artery supplying the affected lobe, thus sparing the contralateral lobe. In some patients with bilobar disease, the first application was performed on the diseased dominant lobe, followed 4–6 weeks later by microsphere application on the untreated contralateral lobe. The location of the catheter during treatment delivery remained consistent with the position defined in the MAA simulation, allowing further post-treatment dosimetric evaluations to be performed. Any adjustments or changes in catheter position between the initial angiogram and therapy were evaluated with repeat MAA scans prior to successful delivery of the microspheres.

Within twenty-four hours after TARE, all patients underwent a post-therapy imaging session also known as “Bremsstrahlung Imaging” consisting of whole-body planar acquisition (supine position, 18 cm/min scans, low-energy, high-resolution collimators, window, 110 +/− 44 keV, zoom = 1, 1024 × 256 pixel matrix) and SPECT/CT images (64 projections, 25 s/proj) on the same scanner (Siemens Symbia Intevo 6, Siemens Healthcare GmbH, Erlangen, Germany). The CT images were reconstructed using 3D-OSEM (5 iterations, 8 subsets) with a Butterworth filter (cutoff, 0.5 cycles/cm; order, 10) and CT-based attenuation correction on a Syngo workstation (Siemens Healthcare GmbH, Erlangen, Germany).

### 2.4. Dosimetric Assessment

Prior to dosimetric evaluations, [^99m^Tc]MAA SPECT images from the initial workup and post-therapy imaging ([^90^Y]-Bremsstrahlung SPECT) were visually analyzed to verify dose delivery and microsphere distribution and to check congruence between previously identified lesions and regions showing the highest uptake of [^99m^Tc]MAA and [^90^Y] microspheres.

For an accurate dosimetric assessment, we used a multi-compartment so-called partition model to estimate the absorbed dose in the tumors. Unlike its predecessor, this model accounts for differences in tracer accumulation (meaning perfusion) between a tumor compartment and the surrounding healthy tissue. As a result, this model is more faithful to the physiological variations in perfusion that can be observed in HCC—the previous model, on the contrary, postulated a homogeneous particle distribution for all hepatic compartments [25,26,27,28,32,33]. As a consequence, absorbed doses are different between tumor and surrounding tissues, and, generally speaking, lower for healthy parts of the liver.

The quantitative uptake analysis consisted of three main steps: (1) registration of the enhanced CT and [^99m^Tc]MAA SPECT/CT images, (2) segmentation, and (3) dose calculation using PMOD software (version 3.8; PMOD Technologies Ltd., Zurich, Switzerland). Registration was performed using data from a contrast-enhanced CT diagnostic scan with [^99m^Tc]MAA SPECT/CT. The volume-of-interest (VOI) segmentation process for the treated liver lobe and tumors within the lobe used an isocontour method. The threshold value of each VOI was adjusted so that the isocontours of the [^99m^Tc]MAA volume of distribution followed the boundaries of the liver and all tumors individually according to Garin et al. [18] and as previously published [17]. The same team of nuclear medicine physicists and a certified nuclear medicine physician inspected all segmented volumes (tumor volumes, non-tumor, or healthy liver volumes).

The activity in the injected healthy liver was obtained by subtracting the total activity from the tumor fraction. Dose calculation was performed using defined VOIs and counts from the [^99m^Tc]MAA SPECT/CT recorded in an MIRD model to provide an estimate of the radiation dose separately for the tumor and normal liver [25,28,32,33]. From these defined VOIs, a target-to-background ratio (TBR) was also calculated. The TBR stands for the amount of activity collected by tumor tissues compared to the activity uptake within the non-tumoral tissue surrounding the lesions. The TBR is an indicator of tumor hyperperfusion to surrounding healthy liver tissue [19]. Figure 2 and Figure 3 show an example of the complete procedure in two patients consisting of target lesion identification, dosimetric analysis, and follow-up after TARE application.

### 2.5. Assessment of Tumor Response and Definition of Liver Decompensation Criteria

Response to treatment was assessed using the modified solid tumor response assessment criteria (mRECIST 1.1) [34,35]. Follow-up examinations included at least one liver scan (enhanced CT or MRI) three months after TARE (following the second lobar treatment, if applicable) and were compared with the multiphase CT scan included in the initial evaluation pre-TARE. As part of the oncology follow-up, liver function, especially albumin and bilirubin levels, were regularly monitored. Treatment-associated adverse events were classified according to the Common Terminology for Adverse Events (CTCAE). Hepatic decompensation was defined as an increase in bilirubin of at least CTCAE grade 3 (at least three times the baseline value or upper limit of normal) or newly developed ascites during follow-up. Patients with ascites but no increase in CTCAE grade 3 bilirubin or with progressive disease (PD) at follow-up were not included in the liver decompensation group as previously published [17].

### 2.6. Ethical Approval

The local ethics committee approved the study (No. EK 392-18) and it complied with the Declaration of Helsinki.

### 2.7. Statistical Analysis

The study was conducted as an observational study. Patient data were analyzed from the pre-TARE planning day (within two weeks before the first TARE treatment). All patients were followed until death or last contact.

Categorical variables (e.g., liver decompensation) were expressed as frequencies and percentages, and continuous variables as mean values and standard deviations. Overall survival was expressed as median with interquartile range (IQR). Statistical differences were determined by the chi-square test or Fisher’s exact test for categorical variables and by the Wilcoxon rank sum test for continuous variables (no Gaussian distribution of the data). In particular, a comparison test without a Wilcoxon distribution was used to compare tumor doses. The tumor threshold dose, which stands for the necessary dose to observe a morphological response, was determined using the receiver operating characteristic (ROC) analysis and the Youden index. OS was defined by the time from treatment to death or last follow-up. Survival rates were assessed using Kaplan–Meier analyses with death as the event. The log-rank test and Cox’s proportional hazards model were used to calculate differences in survival and hazard ratios, respectively.

Univariable and multivariable logistic regression models were used to analyze predictive factors for positive tumor response after therapy. Potential predictive factors were defined according to the univariable analysis and entered into a multivariable bidirectional stepwise regression model. *p*-values <0.05 were considered significant. Statistical analyses were performed using SPSS (version 27.0, IBM, New York, NY, USA).

## 3. Results

Part of the population presented here was the subject of a previous publication [17]. The first dosimetric estimates as well as the biological and anamnestic surveys obtained were completed and integrated into the analysis presented in Table 1. The mean age of the patients at the time of TARE application was 66 ± 13 years, with a majority being males (71%). Seventy-eight patients (74%) had a positive diagnosis of cirrhosis. The major underlying liver diseases diagnosed at the time of enrollment included alcoholic cirrhosis (n = 33; 31%), chronic hepatitis C virus (HCV) (n = 31; 29%), non-alcoholic fatty liver disease (NAFLD, n = 22; 21%), and chronic hepatitis B virus (HBV, n = 16; 15%) infection.

The majority of patients were classified as BCLC B stage (84 patients, 79%). BCLC C stage was reported in eighteen patients (17%). BCLC A disease was present in only four patients (4%; two patients were referred as part of bridging to transplant therapy and two patients were referred for personal convenience after refusing surgery).

Twelve patients (11%) had extrahepatic metastases localized to the lungs (n = 4), lymph nodes (n = 5), adrenal glands (n = 2), and bone (n = 4). The decision for TARE was based on individual decisions discussed and approved by the certified local tumor board for these patients with extrahepatic metastases. Hepatic manifestation was considered the leading prognostic manifestation in all twelve patients. 

Complete or lobar portal vein thrombosis (PVT) was found in 42 patients (40%). More than two-thirds of the patients (n = 76; 72%) were treatment-naive, 23 patients (22%) had undergone preoperative resection, 15 patients (14%) had undergone transarterial chemoembolization (TACE) prior to TARE, and three patients (3%) were referred after percutaneous radiation therapy. Two patients were referred for bridging therapy prior to transplantation (Figure 1).

One hundred and sixty-three TARE applications were performed in 106 patients, and 205 target lesions were defined for response evaluation according to mRECIST 1.1 criteria. Sequential treatment of both lobes was offered to 57 patients in our cohort (54%), and unilobar therapy was performed in 49 patients. In the case of bilateral or bilobar application, the second procedure was repeated within a maximum of 4 to 6 weeks after the first TARE (mean 5.4 ± 0.8 weeks).

The injected activities, hepatopulmonary shunt values, and dosimetric assessments are summarized in Table 1. The average injected activity was 2.3 gigabecquerels (GBq) per [^90^Y] resin microspheres patient. 

### 3.1. Analysis of Response to Treatment

Response to therapy was assessed according to the mRECIST 1.1 criteria. Imaging assessments at three months showed a response rate of 57%; 13 patients achieved complete remission (CR, 12%) and 45% had partial remission (PR, n = 48). These patients are referred to as responders.

In the remaining 45 non-responders, stable disease was observed in 23% of cases (n = 24), and 21 patients (20%) showed radiologic tumor progression at follow-up or died within three months of TARE. There was no significant difference in treatment response between patients with and without hepatic decompensation (Table 1). The overall response rate in patients with PVT was similar at 59% (9% CR, 43% PR, 26% SD, and 22% PD).

### 3.2. Tumor Dose and Prediction of Tumor Response

Initial [^99m^Tc]MAA SPECT/CT images were used for dosimetric calculations for the different TARE applications. Visually, all target lesions showed a greater accumulation of technetium-labeled MAA particles than the surrounding healthy liver tissue. The TBR values, representing the ratio of activity present in tumor tissue to that accumulated in surrounding healthy liver tissue, were measured to be 4.8 ± 2.1. There was also good visual concordance of these tumor accumulations on control [^90^Y]Bremsstrahlung SPECT images after the application of TARE, although the overall measured TBR values were lower at 2.4 ± 1.9 (*p* < 0.001).

For the 205 monitored target lesions, the mean dosimetry for all lesions was 147.4 ± 68.5 Gy. A ROC curve analysis of tumor response allowed us to define an optimal threshold for predicting a positive tumor response after treatment. The maximum value of the Youden index in our series was 135.6 Gy. The associated sensitivity was 84.2%, specificity 89.4%, positive predictive value (PPV) 91.8%, and negative predictive value (NPV) 80.0%.

For lesions treated with a dose greater than 135.6 Gy, 101 lesions responded to treatment (22 CR lesions and 79 PR lesions), while 76 of the remaining 95 lesions below the threshold were non-responders to TARE (43 SD lesions and 33 PD lesions). The mean values in both groups were 148.2 ± 84.5 Gy for responding lesions and 92.8 ± 48.3 Gy for non-responding lesions (*p* < 0.001). Overall, non-responding lesions were larger in diameter than other lesions (8.9 ± 5.6 cm vs. 5.2 ± 4.1 cm, *p* = 0.120) and usually showed more heterogeneous scintigraphic tracer uptake, with frequent necrotic areas on imaging.

A total of 9 lesions treated with a dose greater than 135.6 Gy failed to respond to therapy. These lesions were visually very heterogeneous and showed remaining hyper-perfused areas three months after TARE, particularly at the periphery of necrotic zones. There were 19 lesions treated with a dose of less than 135.6 Gy that, regardless, responded to therapy (PR for all lesions). These were all lesions less than 5 cm in diameter.

For the entire population, a dose of 6.3 ± 2.4 Gy was delivered to the patients’ lungs during the entire course of treatment. No pulmonary radiation complications were reported. The dose to healthy liver tissue was 35.8 ± 9.7 Gy, with a significant difference between responders and non-responders (31.3 ± 8.2 vs. 45.2 ± 11.8 Gy, *p* = 0.004).

Comparing both groups, the presence of liver cirrhosis, the underlying liver diseases, the tumor stadium, as well as the baseline laboratory values for bilirubin and albumin did not show any association (Table 1). Indicators of hepatic functional reserve (Child–Pugh and Albi scores) did not differ between the two groups. Similarly, tumor volume and size, multifocal involvement, and the presence of PVT were not associated with tumor response. On the other hand, TBR values measured on initial [^99m^Tc]MAA-SPECT-imaging and tumor-modeled dosimetric values were significantly different in both groups. Higher TBRs (*p* < 0.001) and higher tumor-absorbed doses (*p* = 0.009) were found in the responder group (Table 1).

### 3.3. Overall Tolerability and Hepatic Toxicity

No deaths were reported within six weeks of treatment. A total of 28 patients developed clinical signs of hepatic decompensation after TARE. There was no significant difference in the prevalence of decompensation comparing responders to non-responders (21% vs. 33%, *p* = 0.098). Both groups also had a relatively similar prevalence of cirrhosis (70% vs. 87%, *p* = 0.298). 

Liver function in HCC patients with hepatic decompensation deteriorated in the three months following TARE, while liver function in HCC patients without hepatic decompensation remained stable. Consistent with the first published results [17] for part of the series, patients with hepatic decompensation had a higher tumor volume (374 ± 84 mL vs. 304 ± 68 mL, *p* = 0.008), as well as higher bilirubin levels (0.7 ± 0.8 mg/dL vs. 1.4 ± 1.2 mg/dL, *p* = 0.022) and lower albumin levels (3.9 ± 0.8 g/dL vs. 3.3 ± 1.2 g/dL, *p* = 0.015). The mean activity applied during TARE was not significantly different (2.1 ± 0.5 GBq vs. 2.6 ± 0.9 GBq, *p* = 0.241). However, the irradiation of healthy liver tissue was lower in patients without hepatic decompensation (32.1 ± 7.9 Gy vs. 46.3 ± 15.8 Gy, *p* = 0.013).

Hepatic decompensation after TARE is likely to have a significant impact on subsequent patient management and overall survival. Eight patients died within three months after TARE. Regardless of the cause of death, these patients were considered to have PD. In general, the occurrence of hepatic decompensation after TARE affected the subsequent management of patients. Of the 28 patients with hepatic decompensation, only 7 (25%) received further HCC treatment after TARE during follow-up, compared to 82% of patients (n = 64) without decompensation (*p* = 0.004).

### 3.4. Overall Survival of Patients after TARE

Kaplan–Meier analysis was performed and survival curves are shown in Figure 4 for both responders and non-responders. Median follow-up was 22.3 months (range of 6–36 months). Median survival was 18.2 months [IQR 10.2–27.4] with significant differences between responders and non-responders (respectively, 24.1 [IQR 13.1–36.4] vs. 10.4 [IQR 6.3–15.9] in months; *p* = 0.022).

Subanalysis of the subset of patients with hepatic decompensation at follow-up showed that prognosis was significantly influenced by the presence of decompensation. Median survival for patients with hepatic decompensation was 4.7 months [IQR 3.1–6.2]. Patients without hepatic decompensation had a median OS of 20.2 months [IQR 14.3–33.2], *p* < 0.001.

## 4. Discussion

### 4.1. Dosimetric Estimation Using [^99m^Tc]MAA SPECT Imaging

In the case of TARE, angiography and [^99m^Tc]MAA SPECT imaging are mandatory steps—on the one hand, for the evaluation of anatomical vascular accesses, and, on the other hand, for the distribution of [^99m^Tc]MAA, which mimics the distribution of [^90^Y] microspheres, which is used for the dosimetric assessment and measurement of liver–lung shunting. From the obtained SPECT images, application of the partition model presented by Ho et al. allows dosimetric determination based on tracer accumulation in tumor tissue and surrounding healthy tissue [28]. This method is considered the most accurate approach, but requires delineating tumor zones and segmenting healthy liver tissue. These steps have already been discussed regarding their robustness and reproducibility, but many authors agree that a strong correlation exists between MAA-based and [^90^Y]-based dosimetry for resin microspheres [36,37,38]. In our cohort, given the frequent presence of lesions larger than 5 cm with necrotic areas, we chose to use a semi-automatic thresholding method based on SPECT images, as described in particular by Garin et al. [18]. This choice, which de facto excludes necrotic areas during the dosimetric estimation steps, leads to a relative overestimation of TBR and dosimetric values compared to manual methods that include necrotic areas. On the other hand, the latter are more difficult to reproduce and lead to greater heterogeneity in radiation assessment [39].

In our 106 HCC patients, measured TBR values on [^99m^Tc]MAA SPECT imaging are close to 5, reflecting significant tumor perfusion compared to surrounding healthy tissue. This condition is a prerequisite that is often cited as an indication for TARE [19,23,24]. High TBR values were also found in our 205 lesions examined on post-therapy [^90^Y]Bremsstrahlung imaging, which seems to support using [^99m^Tc]MAA for therapy planning. However, the mean values found on Bremsstrahlung imaging are lower than those measured by [^99m^Tc]MAA as published by Ilhan et al. [39]. Explanations may include partial volume effects associated with altered spatial resolution or particle size differences [40].

Post-therapeutic dosimetry can be more accurately determined using post-radioembolization [^90^Y] PET/CT scans [30,41,42]. These post-treatment measurements were performed in only five patients from our cohort and showed excellent visual concordance with [^99m^Tc]MAA tracer distribution as published [43,44], but could not be repeated in all patients, mainly due to availability of imaging equipment.

### 4.2. Dosimetry and Prediction of Tumor Response

In our series, we have demonstrated a strong correlation between tumor response and the dose absorbed by the tumor tissue. Predicting this tumor response before TARE is also possible by defining a threshold radiation dose for the lesions. Analysis and follow-up of 205 lesions according to the mRECIST 1.1 criteria allowed us to define a threshold of 135.6 Gy. This threshold is slightly higher than that recommended by Levillain et al. or Garin et al. with a published tumoricidal dose range between 100 and 120 Gy using [^90^Y] resin microspheres [15,19]. This is, at least, partially explained by our choice to exclude necrotic areas from our tumor volumes—leading to higher mean activities in the remaining tumor compartment.

A total of 110 lesions were irradiated with a dose higher than the defined threshold and 101 of them responded, so that a PPV of 91.8% was achieved. This confirms, as other authors have pointed out, the need to achieve a minimum dose absorbed by the tumor in order to obtain good TARE efficacy [4,45,46,47]. Only nine lesions treated with a dose greater than 135.6 Gy failed to respond to therapy. These lesions were all larger than 5 cm in diameter and contained one or more necrotic areas excluded during segmentation. The consequence of this heterogeneity stated for lesions larger than 5 cm is the presence of high-dose tumor areas and other subthreshold areas, as far as can be judged by visual analysis within our cohort. This is evidenced by our observation of tumor recurrence areas found almost exclusively in the periphery of these lesions, probably due to inadequate irradiation in these areas.

Interestingly, responding patients in our cohort were also characterized by less healthy liver tissue irradiation compared to non-responding patients (31.3 ± 8.2 Gy vs. 45.2 ± 11.8 Gy, *p* = 0.004). This finding may seem surprising at first glance, especially considering the higher tumor dose received by responders (148.2 ± 84.5 Gy vs. 31.3 ± 8.2 Gy, *p* = 0.009). A large part of the explanation probably lies in the tumor perfusion itself: responding patients had higher overall TBR values compared to non-responders (6.7 ± 2.3 vs. 2.5 ± 4.1, *p* < 0.001). The higher uptake of [^90^Y] microspheres by the former explains the lower irradiation of healthy residual liver tissue and the less frequent occurrence of hepatic decompensation (21% vs. 33%, *p* = 0.098).

To further quantify the association of this tumor response with other clinical and biological parameters, particularly those characterizing hepatic functional reserve, univariate and multivariate analyses are presented in Table 2. In univariate regression models, only a higher tumor dose and higher TBR (assessed by [^99m^Tc]MAA SPECT) emerged as “predictive factors” for tumor response after TARE. Bilirubin, albumin, Child–Pugh, or ALBI scores were not statistically associated with tumor response. Neither were multifocal tumor involvement or the occurrence of hepatic decompensation during follow-up. In the multivariable logistic regression model, both a dose delivered to the tumor of more than 136.5 Gy and a high TBR were found to be independent predictive factors for tumor response (odds ratio (OR) 4.11 [confidence interval (CI) 1.56–10.42], *p* = 0.034, and OR 4.63 [CI 1.23–12.45], *p* = 0.009, respectively).

### 4.3. Overall Survival after TARE

In our study, the tumor-absorbed dose not only determines the tumor response, but was also associated with patient survival. Responders had higher tumor-tissue-absorbed dose (148.2 ± 84.5 Gy) than non-responders (92.8 ± 48.3, *p* = 0.009) and showed an associated better survival (*p* = 0.022).

To estimate the hazard ratio (HR) for the endpoint of death associated with the different variables likely to influence survival in patients with HCC, a complementary Cox regression analysis was performed (Table 3). In particular, the latter confirmed our first published data [17] regarding the predominant role of hepatic decompensation as a highly relevant risk factor for death (HR: 5.71 [CI 2.72–10.43], *p* < 0.001). Hepatic decompensation is not directly related to the activities administered to the patients, as we have previously shown [17], but the risk of hepatic decompensation increases with the irradiation of the healthy liver parenchyma. This increased non-tumoral compartment irradiation is seen in our population of non-responders (45.2 ± 11.8 Gy vs. 31.3 ± 8.2 Gy, *p* = 0.004).

After adjustment for other variables, younger age and ALBI grade 1 were also found to be independent negative prognostic factors, with HRs of 1.09 [CI 1.04–1.16], *p* = 0.038 and 2.82 [CI 1.43–4.47], *p* = 0.011, respectively.

### 4.4. Hepatic Decompensation

We defined hepatic decompensation as an increase in bilirubin of at least CTCAE grade 3 or the appearance of ascites within three months of TARE application. This definition is intentionally broader than that of radioembolization-induced liver disease (REILD), which has been described as a specific syndrome occurring in patients 4 to 8 weeks after treatment [48]. These differences have been discussed previously [17]. In all cases, the occurrence of serious complications such as hepatic decompensation will have a significant impact on patient prognosis [17]. Liver failure was found in all eight patients who died within three months of TARE, and only 25% of patients with decompensation were offered other therapies during follow-up, compared to 82% of patients without decompensation (*p* = 0.004). Overall survival was also significantly impaired in patients with hepatic decompensation (4.7 months [IQR 3.1–6.2] vs. 20.2 months [IQR 14.3–33.2], *p* < 0.001). These patients had a larger tumor volume (*p* = 0.008), higher total bilirubin levels (*p* = 0.022), and lower albumin levels (*p* = 0.015), probably reflecting impaired liver reserves.

As previously suggested, irradiation of healthy liver tissue appears to play an essential role in the onset of decompensation [17]. Patients with decompensation received a mean radiation dose of 46.3 ± 15.8 Gy compared to 32.1 ± 7.9 Gy for patients without hepatic decompensation—i.e., a statistically higher irradiation of healthy tissue (*p* = 0.013) for comparable [^90^Y] microsphere activities applied during TARE. As previously proposed, these dosimetric values were obtained from initial [^99m^Tc]MAA SPECT [17,19,24,25]. This higher radiation dose in combination with impaired liver function is sufficient to induce hepatic decompensation in HCC patients [49]. Our dosimetric results are in line with the recommendations published by Levillain et al., who defined an upper limit for healthy liver irradiation of 40 Gy for patients with normal liver function and of the order of 30 Gy for patients with impaired liver function [19].

### 4.5. Limitations of This Study

The study is monocentric and retrospective. The patients referred to our clinic for TARE reflect the clinical reality of the proposed therapies, but the characteristics of the patients and the tumors treated are different. For example, only four BCLC A patients were included, and BCLC B tumors were overwhelmingly represented.

Despite these limitations, this study found a strong correlation between tumor dosimetry and response to therapy. Similarly, the occurrence of hepatic complications was shown to be related to the extent of irradiation of healthy liver tissue as previously published [17]. However, these results need to be confirmed by larger randomized trials, in particular, to better assess hepatic tolerance to radiation.

From a technical point of view, the partitioned dosimetric model used in this study allowed only an assessment of the mean dose to each tissue compartment. This approach is limited in large, highly heterogeneous lesions. Implementing voxel-based dosimetry models should make it possible to refine these dosimetric considerations [50].

## 5. Conclusions

This study, as well as previous studies [8,11,14,17,18,29,36,46], demonstrates the predictive value of [^99m^Tc]MAA SPECT imaging in conjunction with dosimetry modeling for both response to radiotherapy and OS. The use of dosimetric models based on data obtained after diagnostic angiography but before TARE is the only way to adapt the therapeutic planning and the activity to be administered. This will be particularly useful for large lesions with a high degree of perfusion heterogeneity—in our series, these lesions larger than 5cm gave the least satisfactory TARE responses.

Our results also confirm that TARE is well-tolerated, despite the fact that almost a quarter of the patients suffered from hepatic decompensation during follow-up. Hepatic decompensation was clearly associated with a higher dose delivered to healthy liver tissue and should be avoidable if dosimetric planning is in place. Randomized trials are needed to confirm the results of this study concerning this point. Finally, our data are in support of the value of TARE in the presence of PVT—as PVT did not have a significant impact on tumor response and was not systematically associated with liver decompensation.

## Figures and Tables

**Figure 1 diagnostics-13-02432-f001:**
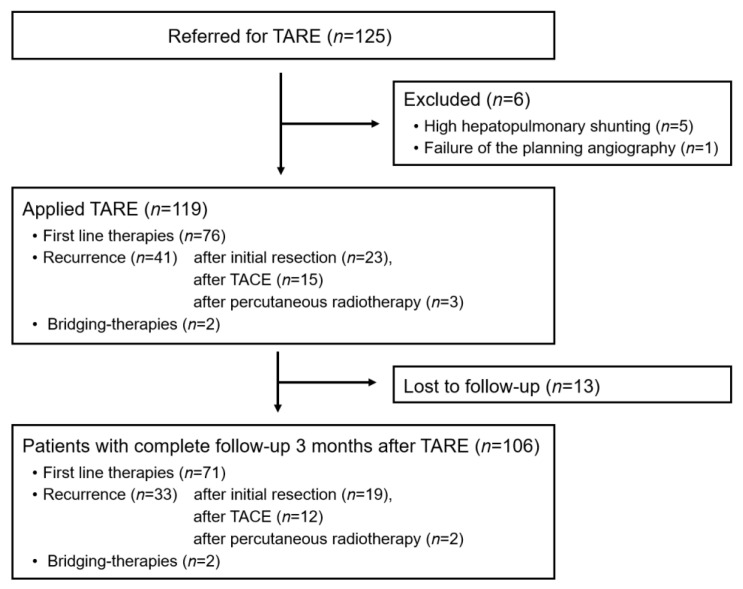
Flow chart of the included patients. Among the 125 patients referred for TARE between January 2015 and February 2023, the therapy could be applied by 119 patients; 13 patients were lost in follow-up.

**Figure 2 diagnostics-13-02432-f002:**
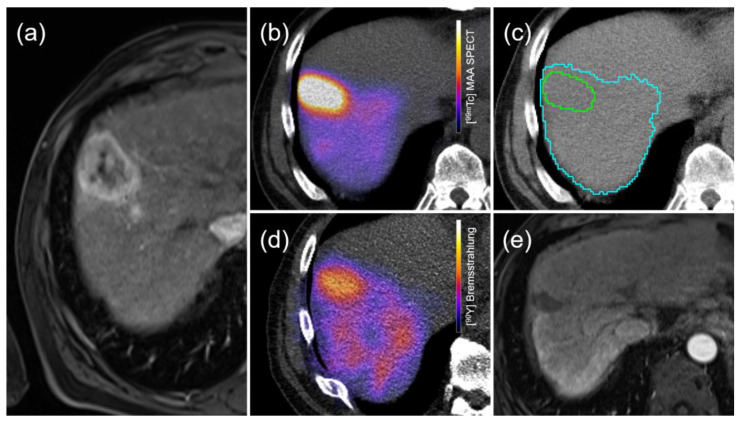
TARE was performed in a 73-year-old patient with multiple HCC lesions in the right liver. (**a**) After intravenous bolus administration of Multihance (Bracco Imaging Deutschland GmbH, Konstanz, Germany), the largest liver lesion (43 × 33 mm) was subcapsular in segment VIII and showed pathognomonic contrast on MRI T1-weighted gradient echo (VIBE) images. The same lesion showed a strong arterial contrast enhancement on the workup CT scan pre-TARE. (**b**) After the angiographic workup and the injection of 105 MBq of [^99m^Tc]MAA into the right hepatic artery, the SPECT/CT images showed a predominant tracer deposition in this lesion. The surrounding tissues showed comparatively little [^99m^Tc]MAA-enhancing activity. (**c**) Segmentation of tumor compartments (cyan isocontour from [^99m^Tc]MAA SPECT overlayed on CT) and healthy tissue (green) was performed from the MAA SPECT/CT images. A TBR of 6.2 was measured for the subcapsular lesion. (**d**) TARE was performed 7 days later. After precise repositioning of the catheter, a total of 1.15 GBq of [^90^Y] microspheres (Sirtex Medical Europe GmbH, Bonn, Germany) was selectively perfused into the right hepatic artery. Compared to the workup session, a similar microsphere distribution with a measured TBR of 3.9 was observed. Dosimetric evaluation using the partition model showed higher tumor irradiation than healthy parenchyma (197 Gy vs. 23 Gy). (**e**) Follow-up MR imaging one year after TARE also demonstrated regression in size and contrast enhancement of the treated lesion. The left lobe of the liver developed a hypertrophy and the patient did not show any signs of hepatic decompensation during the follow-up period.

**Figure 3 diagnostics-13-02432-f003:**
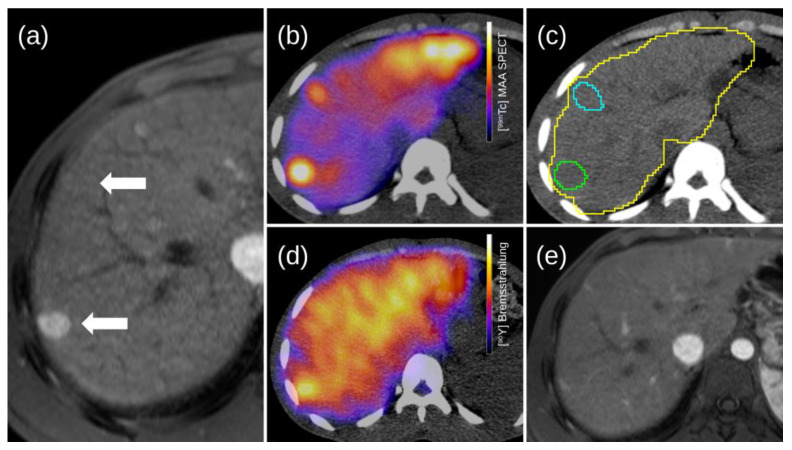
TARE was proposed in a 66-year-old patient with an advanced multifocal bilateral HCC, with six identifiable right lesions and diffuse left liver infiltration. The MRI performed as part of the work up showed multiple hyperperfused areas after the intravenous bolus administration of Multihance (Bracco Imaging Deutschland GmbH, Konstanz, Germany). After coiling the gastroduodenal artery (absence of the cystic artery due to previous cystectomy), the injection was made at the level of the proper hepatic artery (**a**). The arrows indicate two hypervascularized areas of the right liver (segments VII and VIII) on T1-weighted gradient-echo MRI (VIBE) images. These are also seen on SPECT/CT images after injection of 155 MBq [^99m^Tc]MAA (**b**). Segment VII (green isocontour) and VIII (cyan isocontour) with TBR measured at 5.2 and 3.3, respectively, are shown for the two right lesions, the yellow isocontour corresponds to the liver contours (**c**). A new angiography with an activity of 2.16 GBq of microspheres [^90^Y] (Sirtex Medical Europe GmbH, Bonn, Germany) was performed 7 days later, allowing TARE to be applied to the proper hepatic artery. Compared with the workup session, the TBRs measured for the two right lesions shown are lower, at 3.5 and 2.8, respectively (**d**). During follow-up, there was a decrease in size and contrast enhancement of the treated lesions on the right, as well as a normalization of the left perfusion 6 months after the application of TARE (**e**). No new lesions were found on the follow-up MRI scan.

**Figure 4 diagnostics-13-02432-f004:**
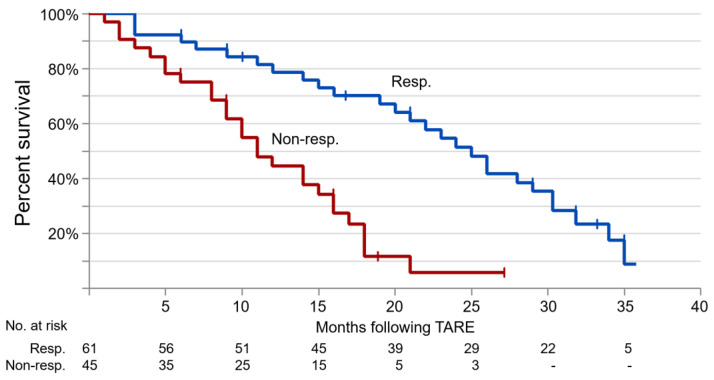
Kaplan–Meier curves for OS in months of HCC patients stratified by TARE response. The median OS is 24.1 months [IQR 13.1–36.4] for patients who respond to therapy (Resp.) and 10.4 months [IQR 6.3–15.9] for patients who do not respond to therapy (Non-Resp.). The numbers at the bottom of the curve indicate patients at risk (No. at risk).

**Table 1 diagnostics-13-02432-t001:** Baseline data and characteristics of enrolled patients *.

	All Patients	Responders	Non-Responders	*p*-Value *
	(*n* = 106)	(*n* = 61)	(*n* = 45)	
Population				
Age (years)	66 ± 13	65 ± 12	67 ± 15	0.789
Male/Female	75 (71%)/31 (29%)	42 (69%)/19 (31%)	29 (64%)/16 (36%)	0.726
Liver cirrhosis	78 (74%)	43 (70%)	39 (87%)	0.298
Underlying liver disease				
Ethylic cirrhosis	33 (31%)	17 (28%)	16 (35%)	0.452
NAFLD	22 (21%)	13 (21%)	9 (20%)	0.521
HCV	31 (29%)	18 (29%)	13 (29%)	0.285
HBV	16 (15%)	12 (20%)	4 (9%)	0.98
Other	4 (4%)	1 (2%)	3 (7%)	0.542
Tumor characteristics				
BCLC stadium				
BCLC A	4 (4%)	3 (5%)	1 (2%)	1
BCLC B	84 (79%)	49 (80%)	35 (78%)	0.201
BCLC C	18 (17%)	9 (15%)	9 (20%)	0.478
Tumor volume (mL)	330 ± 113	294 ± 94	370 ± 196	0.074
Multifocal tumor	88 (83%)	54 (88%)	34 (76%)	0.182
Mean tumor diameter (cm)	6.7 ± 4.8	5.2 ± 4.1	8.9 ± 5.6	0.125
PVT	42 (40%)	15 (25%)	27 (60%)	0.089
Extrahepatic metastasis *	12 (11%)	8 (13%)	4 (9%)	0.235
Baseline laboratory				
Bilirubin (mg/dL)	0.9 ± 0.7	0.7 ± 0.6	1.1 ± 0.8	0.287
Albumin (g/dL)	3.8 ± 0.6	3.9 ± 0.5	3.1 ± 1.9	0.458
Liver functional reserve test				
Child–Pugh score				
Class A	90 (85%)	54 (89%)	36 (80%)	0.352
Class B	16 (15%)	7 (11%)	9 (20%)	
ALBI score				
Grade 1	51 (48%)	34 (56%)	17 (38%)	0.116
Grade 2	55 (52%)	27 (44%)	28 (62%)	
Angiography and MAA SPECT				
Hepatopulm. shunting (%)	6.1 ± 1.7	5.5 ± 1.5	6.8 ± 2.9	0.925
TBR ([^99m^Tc]MAA SPECT)	4.8 ± 3.1	6.7 ± 2.3	2.5 ± 4.1	<0.001
TARE application				
TBR ([^90^Y]Bremsstrahlung)	2.4 ± 1.9	3.1 ± 1.4	1.6 ± 3.2	0.084
Applied activity (GBq)	2.3 ± 0.9	2.6 ± 0.7	2.1 ± 1.7	0.541
Both-lobe appl. (seq. appl.)	57 (54%)	31 (51%)	26 (58%)	0.884
Single-lobe application	49 (46%)	30 (49%)	19 (42%)	0.742
TARE dosimetry				
Lung (Gy)	6.3 ± 2.4	6.1 ± 1.4	6.9 ± 3.2	0.189
Tumor (Gy)	147.4 ± 68.5	148.2 ± 84.5	92.8 ± 48.3	0.009
Non-tumor (Gy)	35.8 ± 9.7	31.3 ± 8.2	45.2 ± 11.8	0.004
Response to therapy				
CR	13 (12%)	13 (21%)		
PR	48 (45%)	48 (79%)		
SD	24 (23%)		24 (53%)	
PD	21 (20%)		21 (47%)	
Hepatic decompensation	28 (26%)	13 (21%)	15 (33%)	0.098
Overall survival (Mo)	18.2 [IQR 10.2–27.4]	24.1 [IQR 13.1–36.4]	10.4 [IQR 6.3–15.9]	0.022

Data are presented as absolute numbers and either percentages or standard deviations. Overall survival is expressed as median with interquartile range (IQR). A *p*-value < 0.05 was considered statistically significant. * Decision of TARE in patients with extrahepatic metastasis was based on individual decisions discussed and approved by the certified local tumor board. In all patients, the hepatic manifestation was considered the leading prognostic manifestation.

**Table 2 diagnostics-13-02432-t002:** Predictive factors for tumor response after TARE.

	Univariable Regression			Multivariable Regression		
Parameters	OR	95% CI	*p*-Value	OR	95% CI	*p*-Value
Dose to tumor > 135.6 Gy	5.73	1.98; 11.56	0.011	4.11	1.56; 10.42	0.034
TBR ([^99m^Tc]MAA SPECT) > 3	6.46	1.42; 11.24	0.005	4.63	1.23; 12.45	0.009
Child–Pugh A	1.93	0.81; 2.95	0.367			
Bilirubin level > 1.2 mg/dL	1.45	0.93; 1.98	0.298			
Albumin level < 3.4 g/dL	1.09	0.91; 1.89	0.487			
ALBI score 1	2.07	0.88; 2.56	0.129			
Multifocal tumor	0.95	0.82; 4.89	0.232			
Hepatic decompensation	0.91	0.76; 2.18	0.134			

Results of the univariate and multivariate regression models of factors predicting tumor response after TARE application. A *p*-value < 0.05 was considered statistically significant. Abbreviations: OR, odds ratio; 95% CI, 95% confidence interval.

**Table 3 diagnostics-13-02432-t003:** Cox regression model for the analysis of the independent risk factors for death after TARE application.

	Cox Single-Variable Regression			Cox Multiple-Variable Regression		
Parameters	HR	95% CI	*p*-Value	HR	95% CI	*p*-Value
Age	1.05	1.02; 1.18	0.024	1.09	1.04; 1.16	0.038
Sex (male vs. female)	0.96	0.43; 1.94	0.934			
Liver cirrhosis	1.31	0.56; 2.91	0.461			
Portal vein thrombosis	1.74	0.83; 4.24	0.217			
Multifocal tumor	1.54	0.91; 3.22	0.354			
Extra hepatic metastasis	1.12	0.56; 2.04	0.425			
BCLC (B vs. C)	1.06	0.38; 3.12	0.823			
Child–Pugh (A vs. B)	1.85	1.14; 6.84	0.038			
ALBI score (grade 1 vs. 2)	2.71	1.32; 4.35	0.002	2.82	1.43; 4.47	0.011
Hepatic decompensation	5.56	2.65; 10.34	<0.001	5.71	2.72; 10.43	<0.001

Results of the univariate and multivariate Cox regression model of risk factor for death after TARE application. A *p*-value < 0.05 was considered statistically significant. Abbreviations: HR, hazard ratio; 95% CI, 95% confidence interval.

## Data Availability

The data supporting this study are not publicly available due to our institution’s ethical and data confidentiality policies and are subject to approval and a data sharing agreement. Please contact our research group at christian.goetz@uniklinik-freiburg.de.

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
