# Peer review of "Predictive Value of [99mTc]-MAA-Based Dosimetry in Hepatocellular Carcinoma Patients Treated with [90Y]-TARE: A Single-Center Experience"

_diagnostics, 2023, doi:10.3390/diagnostics13142432_

Round 1

Reviewer 1 Report

Report of manuscript entitled “Use of Dosimetry based on [99mTc]-MAA SPECT/CT as a predictor of Tumor Response and Survival in Hepatocellular Carcinoma Patients treated with [90Y]-TARE: A single-center experience” by Michael Doppler et al.

Overall comments

The manuscript under review demonstrates the predictive value of [99mTc]MAA SPECT imaging in conjunction with dosimetry modeling for both response to radiotherapy and overall survival (OS). The use of dosimetric models based on data obtained after diagnostic angiography but before TARE is the only way to adapt the therapeutic planning and the activity to be administered. The results also confirm that TARE is well tolerated, despite the fact that almost a quarter of the patients suffered from hepatic decompensation during follow-up. 

I have read the manuscript carefully and I consider it of a medium-level. I think that this work could be of interest for the field of applicative research in TARE and radio-embolization techniques.

The paper is partially clear. The manuscript contains new information. Methodologies can be implemented.

The quantity and the quality of the results presented, in my opinion, is adequate for "Diagnostics". The novelty of this manuscript and its future impact are clear. Therefore, the author could think of a re-submission after Moderate Revision.

Specific Comments

-Title – In my opinion, it is too long and not incisive. 

-The abstract is too long. It must be reduced. it is preferable to avoid too many acronyms. Subdivision into sections is not recommended.

-Keywords – Add “retrospective studies”

-The English should be improved. Some sentences are too long.

-Improving figure 1 in terms of resolution, colors and fonts.

-I suggest adding another figure (after figure 2) from another case study.

-Section 4.3 - This paragraph deserves some further consideration.

Good

Author Response

Dear Colleagues,

Thank you for reviewing our initial submission and we appreciate your comments.

You both requested a revision with the inclusion of additional biological parameters and in particular indicators of hepatic functional reserve. The analysis was completed with the addition of Child-Pugh and ALBI scores for the entire population. A corresponding multivariate analysis was introduced for both tumor response and survival data.

Point-by-point responses (R1),

- The title has been shortened at your request to make it more "incisive".

- Similarly, the abstract has been shortened and the section divisions removed. The use of acronyms has been reduced to a strict minimum - the latter have been retained only in the body of the text.

- the words "retrospective analysis" have been added to the keywords

- The English text has been edited and some sentences shortened to meet your requirements.

- A second illustration of a second patient has been added to our submission.

- The Discussion, and in particular Section 4.3, has been revised to include two new tables and a multivariate analysis.

Point-by-point responses (R2),

- Child-Pugh and ALBI scores have been added to Table 1 in addition to biological data, as requested.

- Multivariate analysis was performed by integrating these indicators of liver function for both tumor response and survival analysis.

We hope that these additions and supplements will meet your expectations and we remain at your disposal for any further questions.

Sincerely,

Christian Goetz, MD, PhD

Reviewer 2 Report

This is an entitled 

 Use of Dosimetry based on [99mTc]-MAA SPECT/CT as a predic-tor of Tumor Response and Survival in Hepatocellular Carci-noma Patients treated with [90Y]-TARE: A single-center experi-ence ,,         by Michael Doppler

In this study, the authors aimed to evaluate whether tumor and healthy liver absorbed dose levels in TARE are predictive of tumor response according to mRECIST 1.1 criteria and overall survival. They showed that 99mTc]MAA SPECT imaging is predictive of response and OS.

This reviewer has some concerns below

1.       Please include the data of liver functional reserve, including ALBI score, Child Pugh score, because it is important factors related to OS

2.       The presented data of baseline laboratory test was quite limited. The authors should add those.

3.       Please showed which factors are correlated with initial 99mTc]MAA SPECT imaging,  dosimetric value and clinical factors , including liver functional reserve

4.        The authors should conduct multivariate analysis regarding the factors associated with treatment response and OS

none

Author Response

(The authors gave the same response as above.)

Round 2

Reviewer 2 Report

This revised manuscript is much improved, while this reviewer had some concerns below

1.

In multivariated anlysis (Table 3) , the authors should include the factors of "Dose to tumor > 135.6 Gy" and "TBR ([99mTc]MAA SPECT) > 3"  and should reanalyze. Those are important factor in this study.

Author Response

We thank the second reviewer for his comments. As has been widely published, the dose delivered to the tumor compartment is a key factor in tumor response to radiation [Lewandowski2005, Garin2011, Garin2012, Levillain2021].

Our complementary analyses show that above 136 Gy, response is significantly improved (OR > 4). Of course, the same is true for tumor perfusion (quantified by TBR), which has a direct influence on radioactive particle deposition and dose delivery (Table 2).

Tumor dose and TBR are, in our opinion, irrelevant for survival analysis. They directly determine tumor response and are discussed with Table 2 for Predictive factors for tumor response after TARE. On the other hand, the dose delivered to the tumor compartment itself has no direct impact on survival or the occurrence of complications (the activities applied to the two patient groups are not significantly different in this population, and the values are in line with those we published previously) - it seems much more relevant to discuss the relationship between the dose delivered to healthy tissue and the occurrence of hepatic decompensation [Reincke2022].

In the case of the analysis of independent risk factors for death after TARE application (Table 3), we have included this hepatic decompensation in addition to the biological values and scores reflecting hepatic reserve. As discussed in Section 4.4, its occurrence depends on the dose delivered to healthy tissue and not directly on the dose delivered to the tumor. Decompensation is the leading cause of death in the multivariate analysis (HR > 5).

We added these remarks to the section 4.3.

Regards,

Christian Goetz, MD, PhD

Round 3

Reviewer 2 Report

The authors responded to my concerns appropriately